# Antimicrobial and Amyloidogenic Activity of Peptides Synthesized on the Basis of the Ribosomal S1 Protein from Thermus Thermophilus

**DOI:** 10.3390/ijms21176382

**Published:** 2020-09-02

**Authors:** Stanislav R. Kurpe, Sergei Yu. Grishin, Alexey K. Surin, Olga M. Selivanova, Roman S. Fadeev, Ylyana F. Dzhus, Elena Yu. Gorbunova, Leila G. Mustaeva, Vyacheslav N. Azev, Oxana V. Galzitskaya

**Affiliations:** 1Institute of Protein Research, Russian Academy of Sciences, 142290 Pushchino, Moscow Region, Russia; st.kurpe@gmail.com (S.R.K.); syugrishin@gmail.com (S.Y.G.); alan@vega.protres.ru (A.K.S.); seliv@vega.protres.ru (O.M.S.); ulya@vega.protres.ru (Y.F.D.); 2The Branch of the Institute of Bioorganic Chemistry, Russian Academy of Sciences, 142290 Pushchino, Moscow Region, Russia; eyugorbunova@rambler.ru (E.Y.G.); mustaeva@rambler.ru (L.G.M.); viatcheslav.azev@bibch.ru (V.N.A.); 3State Research Center for Applied Microbiology and Biotechnology, 142279 Obolensk, Moscow Region, Russia; 4Institute of Theoretical and Experimental Biophysics, Russian Academy of Sciences, 142290 Pushchino, Moscow Region, Russian; fadeevrs@gmail.com

**Keywords:** ribosomal S1 proteins, amyloidogenic regions, toxicity, antibacterial peptides, proteome, mass spectrometry

## Abstract

Controlling the aggregation of vital bacterial proteins could be one of the new research directions and form the basis for the search and development of antibacterial drugs with targeted action. Such approach may be considered as an alternative one to antibiotics. Amyloidogenic regions can, like antibacterial peptides, interact with the “parent” protein, for example, ribosomal S1 protein (specific only for bacteria), and interfere with its functioning. The aim of the work was to search for peptides based on the ribosomal S1 protein from *T. thermophilus*, exhibiting both aggregation and antibacterial properties. The biological system of the response of Gram-negative bacteria *T. thermophilus* to the action of peptides was characterized. Among the seven studied peptides, designed based on the S1 protein sequence, the R23I (modified by the addition of HIV transcription factor fragment for bacterial cell penetration), R23T (modified), and V10I (unmodified) peptides have biological activity that inhibits the growth of *T. thermophilus* cells, that is, they have antimicrobial activity. But, only the R23I peptide had the most pronounced activity comparable with the commercial antibiotics. We have compared the proteome of peptide-treated and intact *T. thermophilus* cells. These important data indicate a decrease in the level of energy metabolism and anabolic processes, including the processes of biosynthesis of proteins and nucleic acids. Under the action of 20 and 50 μg/mL R23I, a decrease in the number of proteins in *T. thermophilus* cells was observed and S1 ribosomal protein was absent. The obtained results are important for understanding the mechanism of amyloidogenic peptides with antimicrobial activity and can be used to develop new and improved analogues.

## 1. Introduction

Antibiotic resistance and amyloidosis are topical problems of modern medicine and biology. Over decades of research, a large amount of research data has been obtained on the causes and mechanisms, however, there are still white spots in the understanding of these two seemingly unrelated phenomena. Antimicrobial peptides (AMPs) are usually amphipathic molecules containing a large number of positively charged and hydrophobic amino acids [1]. The prion domains Sup35 and Ure2, which form amyloid fibrils, are characterized by a high content of polar amino acid residues, such as asparagine, glutamine, tyrosine, and glycine [2]. The facts of the formation of fibrils by antimicrobial peptides and the manifestations of the antimicrobial activity of amyloidogenic proteins indicate the existence of a certain relationship between them [3]. It is known that most antimicrobial peptides (AMPs) may act on the cell wall. Despite the low similarity of AMP and amyloidogenic peptides, the latter exhibit similar activity, leading to cytotoxic effects [3]. It is believed that mature amyloids do not exhibit toxicity [4]. On the other hand, there is increasing evidence that oligomers of amyloidogenic peptides exhibit antimicrobial activity [5]. In any case, it remains unclear how antibacterial activity and the ability to form fibrils are related. Understanding the dependence of the properties of a peptide molecule on its structure is an important component for explaining the nature of this phenomenon. Artificially synthesized peptides can be an ideal model for studying the relationship between antimicrobial activity and the ability to form fibrils.

The feasibility of the task depends on the choice of a target for directed aggregation. The selected protein must be multifunctional and important for the functioning of the bacterial cell. In addition, the selected protein should have its own distinctive properties for a particular organism. A protein that meets these requirements may be the ribosomal S1 protein. This multifunctional protein present only in bacterial cell, being involved in many processes: protein biosynthesis, interacting with mRNA, and initiation of translation [6]. Its knockout leads to cell death, so it can be used as a protein target for aggregation. This protein is unique in its structure. It consists of duplicate S1 domains. The number of domains can vary from one to six, depending on the type of bacteria [7]. For example, the bacterial S1 protein of the Firmicutes type contains 1–4 S1 domains, and the S1 protein of most proteobacteria contains 5–6 domains. In addition, there are unique bacteria such as *Mycoplasma mobile*, their S1 protein consists of only one domain, while performing all its inherent functions. Structurally, these domains are similar to the cold shock domain, which is known to form amyloid fibrils, for the formation of which amyloidogenic regions are responsible. In ribosomal S1 proteins, domains are homologous, but not identical. In unique domains (most differing in various microorganisms), amyloidogenic regions can act as antibacterial peptides, interacting with the “parental” S1 protein (protein of specific bacterial species) and interfering with functioning. This is similar to gene knockout, only at the protein level. In addition, the formed protein aggregates can affect the life of the cell: suppress intracellular transport processes, sorb chaperones, and other proteins. All this, ultimately, can lead to the death of a bacterial cell. The development of new technology and, as a consequence, the creation of antibacterial peptides will be a breakthrough in this field of research. The most obvious advantages of amyloid/antibacterial peptides are: (1) the ability to design targeted peptides; (2) biocompatibility due to their natural origin; (3) resistance to proteolytic degradation due to their aggregation.

Thus, the aim of this work was to study the effect of artificially synthesized peptides based on the ribosomal S1 protein on the cell culture of *T. thermophilus*. To achieve the goal, the following tasks were set: to identify on the basis of the primary structure of the ribosomal S1 protein, regions of the polypeptide chain exhibiting antimicrobial activity and the ability to form fibrils and aggregates; to check the artificially synthesized peptides for antimicrobial activity; to verify the ability of artificially synthesized peptides to form aggregates of proteins and fibrils; to check the toxicity of the studied peptides on eukaryotic cell culture; to investigate the peptides impact on the proteome of *T. thermophilus* cells. If successful, this approach can be considered for pathogenic cells (Figure 1).

## 2. Results and Discussion

### 2.1. Analysis of Amyloidogenic Regions and Tendency for Antimicrobial Activity 

A search for the sequence of amino acids capable of aggregation with the use of the FoldAmyloid [8], Waltz [9], PASTA 2.0 [10], and AGGRESCAN [11] programs showed that all five *T. thermophilus* S1 domains contain amyloidogenic regions (Figure 2). The third and fifth domains are theoretically predicted to be the most amyloidogenic. Twenty potential sites prone to amyloid formation were detected. Interestingly, only two regions, A10A (492–501) and D8A (515–522), were located in the disordered areas and one, L7I (274–280), at the border of the third domains. The remaining amyloidogenic sites were found in the protein domains. The third and fifth domains turned out to be the most amyloidogenic for the S1 protein from *T. thermophilus*.

The most amyloidogenic sequences of the third and fifth domains D9G, E10D, V10I, and V10T were analyzed for compliance with antimicrobial activity. Antimicrobial activity was assessed using four CAMPR3 online server algorithms [12]: support vector machine (SVMs), random forests, artificial neural network (ANN), and discriminant analysis (DA). Of the four predicted amyloidogenic peptides, D9G and V10I were found to be potentially antimicrobial. The HIV-1 TAT protein sequence was added to the N-terminus of V10I and V10T to increase the antimicrobial activity, the ability to penetrate the membrane [13,14,15] and to increase the rigidity and mechanical resistance of amyloid complexes, according to the obtained forecast for the R23I and R23T peptides compared to the unmodified peptides (Table 1).

Figure 2 shows that several programs were used to predict amyloidogenic fragments in the S1 protein sequence. Subsequently, for the synthesis of peptides, the consensus sequences were used. We have synthesized all 8 peptides (Table 1). 

### 2.2. Coagregation of Amyloidogenic Peptides and Ribosomal S1 Protein

Aggregation kinetics studies were carried out for the first time and EM images of preparations of the synthesized peptides based on the S1 protein from *T. thermophilus* were obtained under various conditions: buffer composition, temperature, and incubation time [17]. The selected and synthesized peptides are capable of forming amyloids (Appendix A). The ribosomal S1 protein preparation from *T. thermophilus* under the same conditions formed aggregates that assembled into larger formations [18].

When the R23I peptide is co-incubated with the S1 protein from *T. thermophilus* in 1:1 ratio, both the protein aggregates and fibrils were observed. Moreover, the morphology of fibrils in the mixture of peptide and protein was different from the morphology of peptide fibrils. The fibrils in the mixture were uneven, the surface was covered with fine material. When the R23I preparation was incubated together with the S1 protein in 5:1 ratio, both the protein aggregates and fibrils were observed. The number of fibrils increased, and their morphology was similar to fibrils with the peptide to protein ratio of 1:1.

The R23T peptide after 5 h of incubation at 37 °C under 50 mM TrisHCl, pH 7.5; 150 mM NaCl did not form fibrils, but the aggregates were observed. When coaggregating the R23T peptide with the S1 protein under the same conditions, the aggregates of different sizes with the peptide/protein ratio of 1:1 were observed. Sometimes, in the large aggregates, components of different sizes could be seen. Most likely, with these coaggregation parameters, the peptide and protein began to influence each other. When the peptide and protein were jointly incubated in the ratio of 5:1 (conditions as above), both the fibrils and films were formed. Thin films often merged with the background. One can also see the aggregates significantly differing in morphology and consistency (Appendix A). It was similarly verified that coaggregation of the V10T peptide and the S1 protein at 5:1 ratio led to the formation of fibrils (Appendix A).

Thus, co-incubation of the R23T peptide with the S1 protein led to the formation of fibrils and films, i.e., the components of the mixture affected each other. 

### 2.3. Evaluation of the Aantimicrobial Activity of Synthesized Peptides by Spectrophotometry

Based on the predicted data, the peptides were synthesized and their biological activity was tested on a cell culture of *T. thermophilus*. The initial check (single repeat) of the studied peptides is presented in Figure 3. The effect of kanamycin (commercial antibiotics) on the cell culture growth served as a positive control. It is noticeable that the peptides inhibiting the growth of *T. thermophilus* cells, like kanamycin, have an elongated density graph. It is worth noting that the effectiveness of kanamycin does not exceed 0.8, which is probably a limitation of the measurement method. According to the prediction, peptides exhibiting antibacterial action are R23I, R23T, and V10I. This conclusion was confirmed by repeats of the experiment (5 repeats for R23I and 2 for R23T and V10I each). The minimal inhibitory concentration (MIC) for R23I, R23T, and V10I is 50, 500, and 1000 μg/mL, respectively. The D9G peptide did not show biological activity in the range of studied concentrations. Interestingly, there is a relationship between the effectiveness of the peptide (MIC) and the level of antimicrobial activity predicted by the DA CAMPR3 method (Table 1). The predicted level for the most effective R23I peptide is 0.99, while for the R23T peptide, it lies at the sensitivity limit of 0.59. Thus, an increase in the predicted level corresponds to a decrease in MIC and, accordingly, an increase in the antimicrobial activity of the peptide.

The dose–response relationship was studied for kanamycin (commercial antibiotics), the R23T and R23I peptides (Figure 4). From several mathematical models, the Log-logistic model was chosen as the most reliable for kanamycin, the R23I and R23T peptides (Appendix A). It is noteworthy that for kanamycin, there is a continuous transition from a minimum concentration to a maximum. This is evidenced by the detected concentration close to 50% of the inhibition of cell culture growth (10 μg/mL), while there is no midpoint of the transition on the R23T and R23I response curves. 

The data set from the standard concentrations (1, 10, 50, 100 μg/mL) for the R23I peptide showed close similarity to the mathematical models. For the simulation, a truncated set of concentrations was used, since the R23I peptide at the concentration of 500 μg/mL or more leads to the formation of aggregates in solution and an increase in optical density. Estimating the predictive power of response curve models, we investigated additional concentrations (0.5, 5, 12.5, 20, 40, 80, 160, and 250 μg/mL) and found that at concentrations less than 40 µg/mL, no antibacterial activity occurs. But, when the concentration is reached at 40 μg/mL, a pronounced effect is observed. Observations of this kind are characteristic of an alternative rather than a metric response scale. Such a mismatch is probably related to the mechanism of peptide(s) action. The response curve with the additional data for both kanamycin and the R23T peptide corresponds to the log-logistic model.

Modeling the dose-response relationship demonstrates the correlation between the antimicrobial effect and the concentration of the peptide, which proves the effect of the peptide on inhibition of cell growth.

Another equally important characteristic of antimicrobial activity is the onset of the effect. The most effective R23I peptide was shown to inhibit the growth of cell culture in the first hours of treatment (Figure 5). The DMSO curve reflects the growth of *T. thermophilus* cells under experimental conditions without antimicrobial peptide-treatment. Active cell growth is observed from the initial moment of incubation. The transition to the stationary phase of growth occurs after 6–8 h. The number of bacteria stabilized at 0.8 optical density units. In a medium supplemented with 50 μg/mL kanamycin and the R23I peptide, there was no increase in the optical density, which indicates the absence of bacterial growth.

### 2.4. Morphology of Intact and Peptide-treated T. Thermophilus Cells

Along with the spectrophotometric method for determining the antimicrobial activity of peptides, the viability of cells was evaluated using electron microscopy. It was shown that when the R23I peptide (50 μg/mL) was added to the medium, whole cells were practically absent, the cell fragments (shown in the EM image) and a large number of protein aggregates are often found (Figure 6). With an increase in the concentration of the peptide to 100 μg/mL, compared with the control, the number of cells decreased. The cells often had an uneven surface, cell fragments and many protein aggregates were present. The action of the R23I (1000 μg/mL) peptide did not lead to the morphological changes compared with the control. At the given peptide concentration, cells with a typical morphology and flagella were often found. Many protein aggregates were also observed. The absence of an antimicrobial effect with an increase in the concentration of the peptide to 1000 μg/mL, consistent with the spectrophotometry data of the antimicrobial activity of peptides (Figure 3).

A study with *T. thermophilus* culture showed that a morphology of the cells treated with the R23T peptide at the concentration of 100 μg/mL did not differ from the control (Figure 6G). Protein aggregates and flagella were found. Increasing the concentration to 500 μg/mL led to a toxic effect: cells were often damaged, flagella, a lot of protein aggregates, fibrils in the form of thick long bundles were observed (Figure 6H,I).

The action of the V10I peptide at the concentration of 1000 μg/mL led to cell damage (see Figure 6E,F). The preparation contained protein aggregates and fibrils in the form of bundles of different lengths and diameters. The presence of the V10T peptide in the cell culture had no toxic effect. The cells had normal morphology, “young” cells (small size) were found.

### 2.5. Cytotoxicity of the Studied Peptides 

The results of the human cell viability after incubation with the peptides for 24 h are presented in Figure 7. The peptides have been shown to alter cell viability only when using the concentration of 20 μg/mL and more.

To analyze the cytotoxic and/or cytostatic effect of the cells, after incubation with the peptides (20 μg/mL), they were stained with calcein AM (stains only living cells) and propidium iodide (stains only dead cells). It was shown that all cells after 24 h of incubation with the peptides remained alive.

To study the possible activation of the intracellular proteolytic system in response to the incubation with the peptides, the proteolytic activity of caspase 3 cysteine protease was studied in cells after incubation with the peptides. An insignificant 1.3-fold increase in caspase 3 activity was shown compared to the control conditions.

With the absence of the cytotoxic effect of the studied peptides, the cytostatic effect was analyzed. The analysis was carried out on the basis of a study of the distribution of cells in the phases of the cell cycle, mitotic activity, and expression of nuclear antigen ki-67. The results are given in Table 2. It was shown that the incubation with the peptides resulted in a complete suppression of mitotic activity and accumulation of cells in the G1 phase of the cell cycle. To study the accumulation of cells in the G0 phase, after incubation with the peptides, the expression of nuclear antigen ki-67 was analyzed. It was shown that the incubation of cells with the peptides does not induce a decrease in ki-67 expression.

Next, the effect of peptides on the migration activity of cells was investigated. It was shown that the peptides are capable of significant suppression of the migration activity of cells. The number of migrating cells after incubation with peptides was 15 ± 4% relative to their number in the control culture. In the future, it is of interest to test the effect of peptides at different concentrations (50 μg/mL and 100 μg/mL).

### 2.6. Study of the Proteome of Intact and Peptide-Treated T. Thermophilus Cells

According to the UniProt database, 2201 unique proteins of *T. thermophilus* are known, of which only 376 have a complete description. This indicates that there are difficulties in identifying and studying the proteins of this organism. 

The action of the R23I peptide led to a change in the proteome of *T. thermophilus* cells relative to the control one. A total of 605 unique proteins were determined in *T. thermophilus* cells using a mass spectrometer. In the samples, peptide-treated with 20 and 50 μg/mL R23I, 40% and 48% less numbers of proteins were detected, respectively, relative to the control (Table 3 and Appendix A). The treatment of *T. thermophilus* cells with the R23I peptide at the concentration of 100 μg/mL did not lead to a significant increase or decrease in the number of proteins.

On the other hand, an increase in the number of unique and total proteins reaffirms the fact that using the R23I peptide at the concentration of 100 μg/mL or higher leads to a decrease in the antimicrobial effect on *T. thermophilus* cells.

Comparison of cellular proteins annotated as cytoplasmic and membrane showed that when the culture was treated with the R23I peptide at the concentration of 20 and 50 μg/mL, a decrease in the number of cell cytoplasmic and membrane proteins was observed in comparison with the control (Figure 8). This decrease is likely due to a general decrease in the amount of protein in the samples with exception for membrane proteins in the cells treated with 100 μg/mL R23I. Deviation from the general dynamics may indicate the mechanism of R23I action at the concentration of 100 μg/mL, mainly associated with the cell membrane.

Comparing the number of proteins separated by keywords and the total number of annotated proteins in groups 0, 20, 50, and 100 μg/mL of R23I reveals some differences from the general dynamics in the number of certain proteins. The ratio of 201 proteins from the cells treated with 50 μg/mL R23I to 386 proteins from the intact cells is 0.52. A decrease in this proportion below 0.52 indicates the groups of proteins that are vulnerable to the action of the peptide. By this parameter, 28 keywords were sorted and divided into 4 functional groups (Appendix A).

Under the action of 20 and 50 μg/mL R23I, the amount of aminoacyl-tRNA synthetases and proteins involved in the biosynthesis of amino acids (biosynthesis of isoleucine, threonine, methionine, histidine, and aromatic amino acids) decreases in the cells. Besides reducing biosynthetic activity, there was a decrease in the amount of proteins having proteolytic and nuclease activity. A decrease in the intensity of anabolic and catabolic processes may indicate the presence of R23I treated cells under stressful conditions. Possibly, a reducing biosynthetic activity was determined by a decrease in the level of energy metabolism. Thus, for example, a decrease in amount of glyoxylate pathway enzymes and proteins involved in redox reactions can lead to a decrease in the production of ATP, NADF, and other macroergs. Therefore, reducing anabolic processes of proteins and nucleic acids biosynthesis may be due to the energy loss caused by a violation of the cell wall and/or membrane functional integrity. However, the existence of intracellular targets for the R23I peptide cannot be excluded.

Interestingly, in the samples treated with 20, 50, and 100 μg/mL R23I, the amount of ribosome biogenesis proteins is reduced. which may indicate a decrease in the intensity of protein biosynthesis. It should be noted that the amount of detected ribosomal proteins remained approximately at the same level under the action of R23I at a concentration of 20 and 50 μg/mL and without the peptide treatment. But under the action of 100 mg/mL of R23I, a more than two-fold decrease in the proteins number was detected in a comparison with the control (Table 3). Such reduction does not correspond to the general change in the amount of detected proteins described above (the decrease at 20 and 50 μg/mL R23I, the increase at 100 μg/mL R23I). Besides, NSAF (normalized spectral abundance factor, a method of label-free quantitation [19]) did not reveal statistically significant differences in the amount of ribosomal proteins in the cells incubated with 20 and 50 μg/mL of R23I and without peptide (Pearson’s Chi-squared test *p*-value > 0.05). The divergence between the dynamics of ribosomal proteins and change in the total protein composition under the action of the peptide or without it may indicate the resistance of the protein synthesizing apparatus in cells treated with 20 and 50 μg/mL of R23I. Interestingly, most of the ribosomal proteins in the sample of cells treated with 100 μg/mL of R23I were detected on the border of determination (3 peptides for each protein were detected).

The dependence of protein size on NSAF allows dividing ribosomal and membrane proteins into two groups (Figure 9). In general, the ribosomal proteins, with the exception of S1, are characterized by a wide range of distribution by the NSAF parameter and a narrow range by protein size, while the opposite is true for the membrane proteins. A distinctive feature of the ribosomal S1 protein of *T. thermophilus*, which distinguishes it in the group of ribosomal proteins, is its high molecular weight. The absence of S1 ribosomal protein in the samples of cells treated with 20 and 50 μg/mL of R23I may indicate a target-specific effect of R23I against ribosomal protein (Table 3). Besides, the correspondence of the S1 dynamics to the change in the total protein composition may indicate the sensitivity of this protein to the peptide in the medium. It is interesting to note that under the action of 100 μg/mL of R23I, despite the qualitative and quantitative (NSAF) decrease of ribosomal proteins, the quantitative NSAF parameter of the S1 protein does not change. The L2, L5, L11, L13, and S4 proteins show the same dynamics as the S1 protein. A decrease in the amount of ribosomal proteins may indicate a violation of protein biosynthesis under the action of 100 μg/mL of R23I. The nonlinearity of qualitative and quantitative changes in the proteome composition of *T. thermophilus* cells may indicate the existence of two or more mechanisms of the R23I action. 

A third of the “marker” proteins are associated with redox processes of energy metabolism. In addition to the proteins involved in electron transport, the following are especially notable: cytochrome c oxidase subunit 2, isocitrate dehydrogenase (NADP), NADH-quinone oxidoreductase subunit 2, and redox-sensing transcriptional repressor Rex, which regulates anaerobic metabolism [20]. This repressor leads to anaerobic regeneration of NAD+, through the formation of lactate, formate, ethanol, nitrate respiration, and ATP synthesis, which may confirm the hypothesis of a decrease in the production of ATP, NADF, and other macroergs.

The presence of the R23I peptide induces a stress response in *T. thermophilus* cells. This is evidenced by the detection of proteins such as precorrin-6Y C(5,15)-methyltransferase (decarboxylating), which is involved in the stress response of thermophilic bacteria to heat shock [21]. Phosphate import ATP-binding protein (PstB) regulates the intake of polyphosphates, which are necessary in protective reactions under conditions of proteolytic oxidative stress. It is also known that polyphosphates are able to exhibit chaperone functions [22]. The possible key role of polyphosphates in the stress response to the action of R23I can be indicated by the presence in the samples, in addition to PstB, of an enzyme that regulates the production of exopolyphosphatase polyphosphates. The presence of pilN protein may indicate the absorption of exogenous DNA molecules by *T. thermophilus* cells that could be present in the medium due to cell lysis under the action of R23I [23]. Another possible response mechanism may be the removal of the R23I peptide or its modified product from the cell, as indicated by the presence of type II PulD secretin. Of particular interest is the presence of enoyl-ACP reductase, PstB, and metallo-beta-lactamase protein, which determine the resistance of microorganisms to antibiotics [24,25,26].

Thus, at the molecular level, the mechanisms of adaptation to stressful conditions caused by the presence of R23I in the cellular medium are clearly manifested.

## 3. Materials and Methods 

### 3.1. Analysis of Amyloidogenic Regions and Tendency for Antimicrobial Activity

To predict the regions of the ribosomal S1 protein prone to aggregation, four bioinformatics programs were used. A search for the sequence of amino acids capable of aggregation showed that all five S1 domains of *T. thermophilus* contain amyloidogenic regions according to the programs FoldAmyloid [8], Waltz [9], PASTA 2.0 [10], and AGGRESCAN [11].

### 3.2. Chemical Synthesis of Unmodified Peptides and Peptides Modified by HIV Transcription Factor Fragment

The chemical synthesis and purification of the studied peptides (Table 1) is a complex task due to the low solubility of the peptides in water and organic solvents. The procedure of synthesis was described in detail in our previous paper [17]. The purified peptide was tested using an Orbitrap Elite mass spectrometer (Thermo Scientific, Dreieich, Germany). The estimated peptide molecular weight coincided with the calculated one. 

### 3.3. Coagregation of Amyloidogenic Peptides and Ribosomal S1 Protein

#### 3.3.1. Thioflavine T Fluorescence Assay

Fibrils growth was monitored by thioflavine T (ThT) (Sigma-Aldrich, St. Louis, MO, USA) fluorescence. The kinetics of aggregation of peptides was studied using the method of fluorescence spectroscopy. Each kinetics graph was plotted based on the average of several results measured at different time intervals. To facilitate dissolution, all the samples of peptides were pre-dissolved in 100% DMSO, then buffers 50 mM Tris-HCl, pH 7.5, 150 mM NaCl or 20% acetic acid pH 2.0, 150 mM NaCl were added (1% final concentration DMSO) and filtered through 0.22 mm membranes to remove small aggregates. ThT is the most commonly used fluorescent dye for the following amyloid formation semi-quantitatively in vitro. Free ThT has excitation and emission maxima at ∼350 and ∼450 nm, respectively (after binding to fibrils, the excitation and emission λmax change to ∼450 and ∼482–485 nm, respectively). A stock solution of ThT was prepared at a concentration of 20 mM in 50 mM Tris-HCl, pH 7.5, 150 mM NaCl or 20% acetic acid pH 2.0, 150 mM NaCl, respectively and stored at 4 °C protected from light. The final concentration of ThT was 0.2 mM, and each peptide was 0.2 mM. ThT was present in the sample during the fibril formation reaction. Measurements of ThT fluorescence were performed on a Cary Eclipse fluorescence spectrophotometer (Varian, Mulgrave, Australia) in quartz cells 3 × 3 mm at 37 °C.

#### 3.3.2. Transmission Electron Microscopy

The presence of amyloid fibrils in the test samples was monitored by transmission electron microscopy (TEM). End-point solutions of peptides were collected and the samples were dissolved in the same solution to 0.2 mg/mL before studying with electron microscopy. A formvar-coated copper grid 400 mesh (Electron Microscopy Sciences, Hatfield, PA, USA) was placed on a 10 μL sample. After 5 min absorption, the grids with the preparation were negatively stained for 1.5–2.0 min with 1% (weight/volume) aqueous solution of uranyl acetate. The excess of the staining agent was removed with filter paper. The preparations were analyzed using a JEM-100C (Jeol, Tokyo, Japan) transmission electron microscope at the accelerating voltage of 80 kV. Images were recorded on the Kodak electron image film (SO-163) at nominal magnification of 40,000–60,000.

### 3.4. Evaluation of Antimicrobial Activity of Synthesized Peptides by Spectrophotometry

UV-Vis spectra were recorded using a spectrophotometr SF-102 (Akvilon, Moscow, Russia). At the end of the incubation, the density of the medium was estimated by spectrophotometry in a cuvette 10 mm with a total volume of 100 μL. The cuvettes were filled with the peptide water solutions and spectra were collected, using a wavelength from 600 nm. Biological activity of peptides was evaluated by their minimum concentration, which that may inhibit the growth of *T. thermophilus* bacteria culture. Biological activity of peptides was counted by using the following equation:
(1)$$E=1-\frac{A\;\left(sample\;with\;peptide\right)}{A\;\left(sample\;without\;peptide\right)},$$
where *E* is a biological activity of peptides, *A* is an optical absorption after 24 h incubation with and without peptide.

### 3.5. Construction of Response Curves and Evaluation of the Antimicrobial Effect of the Peptide

To construct a dose-effect curve and estimate the quantitative parameters of the curves, the statistical environment R and the drc package were used [27,28]. The choice of model for describing the functional dependence was determined by several parameters: the logarithm of the maximum value of the likelihood function (logLik), the Akaike information criterion (IC), and the residual variance (Res var).

The algorithm for evaluating antimicrobial activity includes the study of this process and the dose–response relationship. The reaction of a biological object determines a quantitative assessment of the antimicrobial effect, depending not only on the concentration of the peptide, but also on the time of its action. Almost any impact begins with a certain threshold. An important final result of the experiment is the assessment of this threshold—the minimum inhibitory concentration (MIC) at which peptide influence is observed.

The type of functional dependence cannot be determined theoretically, however, using modern computer methods, it is possible to select the optimal dose–response curve from experimental data. To describe the response in metric form, nonlinear models are used, which in general terms can be represented as follows:
j (x; b, c, d, e,…) = c + (d − c) φ (x; b, e,…)(2)
where parameters c and d are the lower and upper limits of the response, and φ is some specified nonlinear function with parameters b and e. b is the slope coefficient in the region of the transition state, and e determines the position of the inflection point. The difference (d–c) of the parameters of the dose-effect model determines, in fact, the full effect or the maximum excess of the quantitative response above the normal level. A list of some of the main models is shown in Table 4.

To construct the dose-effect curve and estimate the quantitative parameters of the curves, the statistical environment R and the drc package were used [27,28]. The choice of model for describing the functional dependence was determined by several parameters: the logarithm of the maximum value of the likelihood function (logLik), the Akaike information criterion (IC), and the residual variance (Res var).

### 3.6. Toxicity Determination to Eukaryotic Cells 

We used human skin fibroblasts in our research. Cells was incubated in media alfa MEM (Sigma-Aldrich, St. Louis, MO, USA) with 10% fetal calf serum (Thermo Scientific, Waltham, MA, USA) and 40 mkg/mL gentamicin sulfate (Sigma-Aldrich, St. Louis, MO, USA) at 37 °C in 5% CO_2_.

#### 3.6.1. Cell Isolation Human Skin Fibroblasts

Human skin fibroblasts were isolated from inner surface of the brachia skin sample using explant methods. A skin sample explant (5 × 5 mm) was washed for several minutes in phosphate buffer containing 400 mkg/mL gentamicin sulfates. Tissue was mechanically fragmented in a 1–2 mm^2^ area. After the tissue pieces were attached to the surface of culture dishes, they were allowed to dry in the air for 5–7 min, covered by media alfa MEM with 10% fetal calf serum, and incubated at 37 °C in 5% CO_2_. Media was substituted near skin sample every 3–4 days. Exited from tissue pieces cells were detached from the surface of the culture dishes and counted. In this study we used 1–4 passage cells. 

#### 3.6.2. Cell Viability Analysis

Cell viability was estimated by resazurin cell viability assay. Resazurin at a concentration of 30 μg/mL was added to the cells after 24 h incubation with peptides. Then, the cells were incubated with the dye for 4 h at 37 °C and 5% CO_2_; fluorescence intensity was measured at 595 nm with an Infinity F 200 plate spectrofluorimeter (Tecan, Grödig, Austria). All measurements were carried out on the control samples that were not treated with cytotoxic substances.

#### 3.6.3. Live and Dead Cell Assay

Determine numbers of live and dead cells were carried out by staining cells of the calcein AM (Sigma-Aldrich, St. Louis, MO, USA) and propidium iodide (Sigma-Aldrich, St. Louis, MO, USA). Cells were detached from to the samples surface by Accutase solution (Sigma-Aldrich, St. Louis, MO, USA). Cells were stained in media L-15 (Sigma-Aldrich, St. Louis, MO, USA) containing 1% fetal calf serum, 1 mkg/mL calcein, and 2 mkg/mL propidium iodide for 25 min at 37 °C. An analysis alive and dead cell was conducted by flow cytometer BD Accuri C6 (BD Bioscience, San Jose, CA, USA).

#### 3.6.4. Caspase-3 Activity Analysis

The caspase-3 activity analysis was based on the fluorescence of substrate Ac-DEVD-amc (Enzo Life Sciences, Farmingdale, NY, USA). Cells were incubated for 24 h, then cells were lysed for 10 min in buffer solution contained 25 mM TRIS-HCl (pH 8.0), 60 mM NaCl, 2.5 mM EDTA, and 0.25% NP 40. The fluorescence substrate Ac-DEVD-amc (5 mkM) was added to cell lysates and incubated for 60 min at 37 °C. The caspase-3 activity was analyzed by fluorimetry (360 nm of excitation and 465 nm of emission) in TECAN Infinity F200 plate reader (Tecan, Grödig, Austria).

#### 3.6.5. Cell Cycle Phase Distribution Analysis

To study the distribution of cells by phases of the cell cycle, cells were collected, centrifuged in phosphate-buffered saline (300× *g*, 5 min), fixed with 70% ethanol (24 h, −20 °C). Cells were stained in media containing phosphate-buffer saline, 0.1% Triton ×100, 10 μg/mL propidium iodide, 100 μg/mL RNAase (Biolot, Moscow, Russia) for 15 min at 37 °C. The analysis was performed by flow cytometer BD Accuri C6 (BD Bioscience, San Jose, CA, USA). The results were processed using the ModFit LT 4.1 software (Verity Software House, Topsham, ME, USA).

#### 3.6.6. Analysis of Mitotic Activity

Determination of cells mitotic activity was carried out after 24 and 120 h of cultivating, cells were centrifuged in phosphate-buffer saline (300× *g*, 5 min) and fixed in 70% ethanol (24 h, −20 °C). Number of mitotic cells was counted by fluorescence microscope DM 6000 (Leica, Wetzlar, Germany) after staining by 1 mkg/mL fluorescent dye bisBenzimide H 33342 for 10 min at room temperature. In total, at least 500 cells were analyzed.

#### 3.6.7. Analysis of Ki-67 Expression

To analyze the expression of the Ki-67, cells were collected, centrifuged in phosphate-buffer saline three times (300× *g*, 5 min) and fixed in 70% ethanol (24 h, −20 °C), then the fixed cells were centrifuged thrice (350× *g*, 5 min) in Cell Staining Buffer (Biolegend, San Diego, CA, USA). Cells were stained by antibody PE anti-human Ki-67 (Biolegend, San Diego, CA, USA). Control cells were stained by PE Mouse IgG1, K isotype Ctrl (Biolegend, San Diego, CA, USA) and incubated at room temperature for 30 min at dark. Then the cells were centrifuged twice with a Cell Staining Buffer. Assay was carried out by BD Accuri™ C6 flow cytometer (BD Bioscience, San Jose, CA, USA).

#### 3.6.8. Cell Migration Analysis

The analysis of the migration activity of cells was carried out using Transwell system (Corning Inc, Corning, NY, USA). For analysis, cells were seeded in Transwell, cultured for 24 h, then the culture medium from the lower chamber was replaced with new alfa MEM supplemented with 10% FCS, the culture medium from the upper chamber was replaced with new alfa MEM medium supplemented with 1% FCS, with or without added peptides and incubated for an additional 12 h. The number of cells that migrated to the lower chamber was determined by staining the cells with crystal violet (Sigma-Aldrich, St. Louis, MO, USA).

#### 3.6.9. Statistical Analysis

Data are presented as the mean ± standard deviation (M ± SD). The experiments were performed in at least three repetitions (*n* ≥ 3). The statistical significance of the difference was determined using analysis of variance (ANOVA). 

### 3.7. Mass Spectrometry Analysis 

*T. thermophilus* cells were incubated with or without peptides for 24 h. After measuring the optical density of the cell culture, the experimental and control cell preparations were lysed using an ultrasonic disintegrator SONIFIER 450 (Branson Ultrasonics Corporation, Danbury, CT, USA) processing time 10 min, pulse/pause cycles = 3/3 s, amplitude 35%, power 22.5 W. The cell extract was not allowed to heat above +10 °C. Cell debris and aggregates were collected by sedimentation using centrifugation in a 221.23 V20 rotor of a Z 36 HK centrifuge (Hermle Labortechnik GmbH, Wehingen, Germany) at 20,000× *g* 20 min at +4 °C. The supernatant was dried using a vacuum Concentrator 5301 (Eppendorf, Hamburg, Germany). The prepared lysate was dissolved in 100 μL of 9M urea to denature the total protein. Then, 20 μL of cell lysates was separated by electrophoretic in 12% SDS-PAGE. Protein fractions were divided on three equal parts, that every part was chop and treated with trypsin protease. Mass spectrometry analysis of every part was carried out independently. To purify the samples treated with trypsin, filtration was carried out through ZipTips (MilliporeSigma, Burlington, MA, USA). Protein hydrolyzates were concentrated on a pre-column Acclaim PepMap 100 (C18, 3 μm particle size, 100 Å pore size, 300 μm inner diameter × 5 mm length) and separated on an analytical column Acclaim PepMap RSLC (2 μm particle size, 100 Å pore size, 75 μm inner diameter × 150 mm length) (Thermo Scientific, Waltham, MA, USA) using nanoflow liquid chromatograph EASY nLC 1000 (Thermo Scientific, Waltham, MA, USA). A high-performance mass spectrometer an Orbitrap Elite (Thermo Scientific, Dreieich, Germany) was used as a detector. Panoramic spectra were recorded in the mass range m/z 300–2000 at resolution 60,000; ion fragmentation was performed in an HCD cell; the fragmentation spectra were recorded at resolution 15,000. Preliminary selection of identified peptides with a molecular mass (MM) >600 Da was carried out according to the evaluation criteria of the PEAKS Studio 7.5 program (Bioinformatics Solution Inc., Waterloo, ON N2L 6J2, Canada). The peptides with the threshold function value *T* = −10·log*P* > 15 were considered significant, where *P* is the probability that false identification will reach the same or better match score. The *T* value of >15 corresponds to the *p* criterion <0.03 [29]. The received fragmentation spectra were analyzed by web-server Identipy [30]. Identification of proteins in the mass spectra of the total protein was carried out using the database of the reference *T. thermophilus* proteome (Uniprot, No. UP000217909). Proteome data every part was summarized. Subsequent analysis was conducted by using the statistics software R with packages UniProt.ws, tidyverse, ggplot2, splitstackshape, reshape, readr, RVenn, UpSetR (https://rdrr.io/bioc/UniProt.ws/; https://CRAN.R-project.org/package=readr; https://CRAN.R-project.org/package=splitstackshape; https://CRAN.R-project.org/package=readr; https://CRAN.R-project.org/package=RVen) [31,32].

Annotation of the certain proteins by keywords of the Uniprot database allows us to evaluate the changes in protein composition in 10 categories: biological process, cellular component, variety of coding sequences, molecular function, and others. The number of keywords corresponds to the number of annotated proteins.

## 4. Conclusions

Studying the properties of amyloidogenic peptides with pronounced antimicrobial activity is important for understanding the nature of these compounds. Understanding the dependence between antimicrobial activity and amyloidogenic ability of the peptide will allow us to construct peptides with desired properties. The present work demonstrated the high potential of bioinformatics programs for predicting amyloidogenic ability and antibacterial activity. 

Among the studied peptides, only the R23I peptide proved to have the highest antimicrobial activity comparable with commercial antibiotics. The minimum inhibitory concentration (MIC) of the R23I peptide is 50 μg/mL. With an increase in concentration of more than 500 μg/mL, a decrease in the inhibition of the growth of *T. thermophilus* cells is observed. It is important to note that a decrease in the amount of certain proteins demonstrates a decrease in the antimicrobial activity at the molecular level even with the action of 100 μg/mL. MIC for the R23T and V10I peptides are respectively 500 and 1000 μg/mL, which indicates their lower efficiency compared to the R23I peptide. 

The membranolytic mechanism of the R23I peptide action on the *T. thermophilus* cell culture was suggested. Electron microscopy and mass spectrometry data indicate damage in the membrane, violation of energy metabolism and the secretory system of bacteria. Disruption of the cell membrane morphology under the action of the R23T and V10I peptides indicates a similar mechanism of action. However, a decrease in protein production at a concentration of 20 μg/mL R23I (below MIC) may indicate the presence of another intracellular mechanism of antimicrobial action. Thus, a peptide with both antimicrobial and amyloidogenic properties is possibly able to combine several mechanisms of action depending on its concentration in the medium.

Most of the studied peptides caused a decrease in the mitotic activity of eukaryotic cells, while inhibition of cell culture growth occurred only under the action of the V10I, R23I, and R23T peptides. It was also shown that incubation of human fibroblasts with the studied peptides definitely inhibits the migration activity of cells. In addition, a change in the viability of both eukaryotic and bacterial cells under the action of 20 μg/mL R23I may indicate a non-specific effect of the R23I peptide, and possibly all of the studied peptides. In the future, selected candidate AMP can be modified in such a way as to reduce both the MIC value in relation to the pathogenic organism and the cytotoxic effect in relation to eukaryotic cells [13,33]. The revealed features of the peptide action can be used in the future to develop targeted AMPs for pathogen cells (Figure 1). In our next paper, we will check this suggested approach for *Pseudomonas aeruginosa* cells.

## Figures and Tables

**Figure 1 ijms-21-06382-f001:**
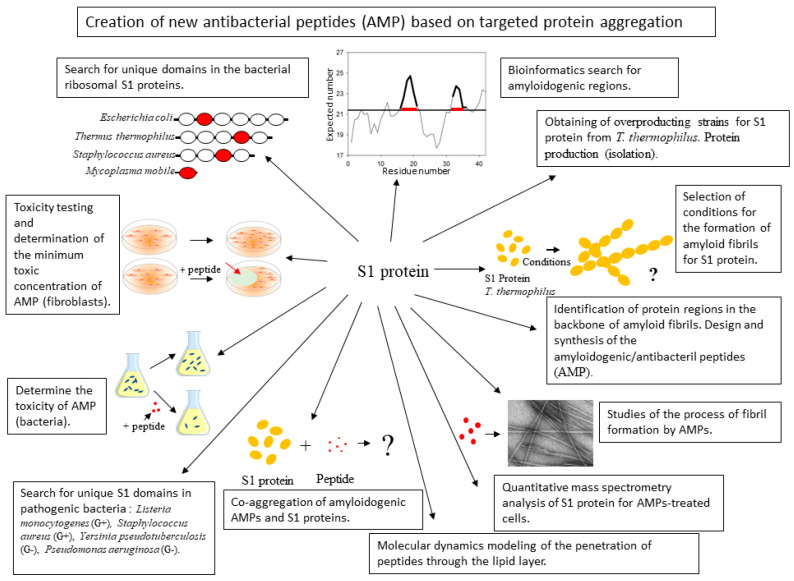
Schematic representation of the main challenges associated with the development of new antibacterial peptides.

**Figure 2 ijms-21-06382-f002:**
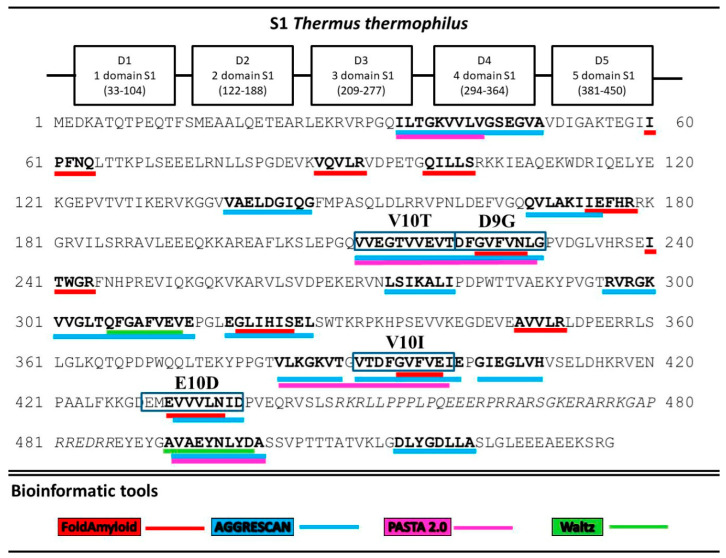
Prediction of amyloidogenic regions prone to aggregation. Amyloidogenic fragments for synthesis are outlined in the rectangles.

**Figure 3 ijms-21-06382-f003:**
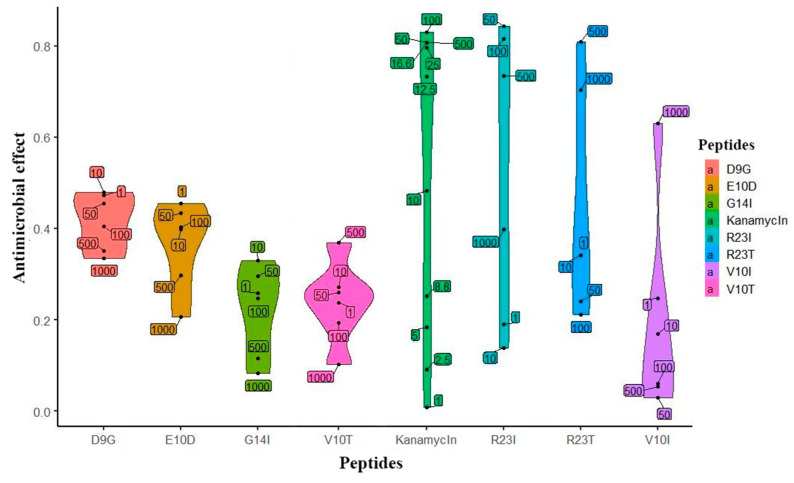
Violin plot of the effect distribution versus peptide concentration. Peptides are located on the abscissa, and the effect is on the ordinate. The upper and lower borders on shapes are show the 95% confidence intervals. The width of the figure shows the frequency of observation of points. Numeric labels with black dots in the plot area reflect the actual concentration of the peptide (μg/mL).

**Figure 4 ijms-21-06382-f004:**
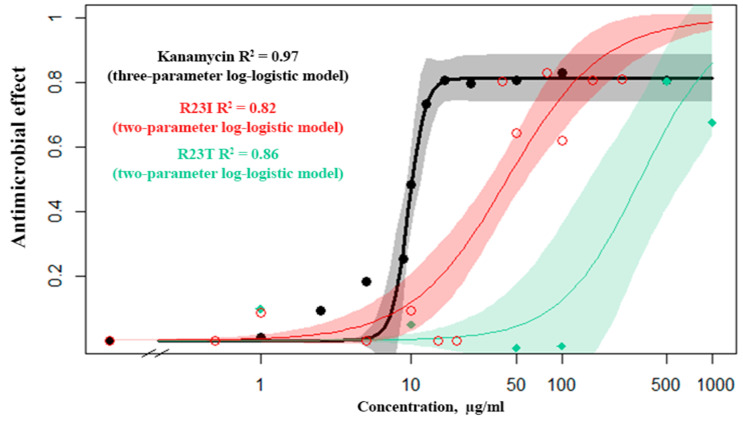
Response curves of *T. thermophilus* cells to the action of kanamycin (black), R23I (red), R23T (green) peptides.

**Figure 5 ijms-21-06382-f005:**
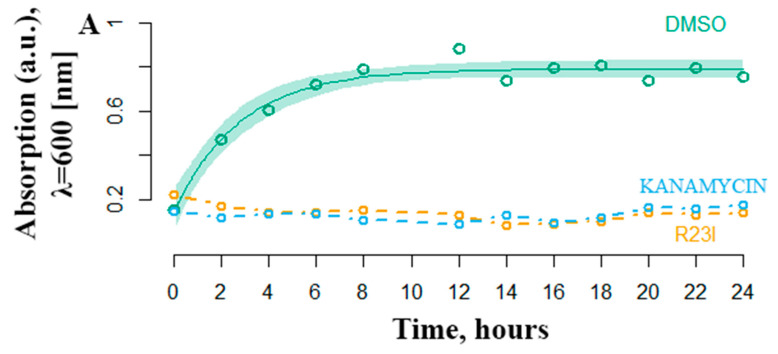
Temporal dependence of changes in the optical density of *T. thermophilus* cells with the addition of kanamycin (commercial antibiotics) and the R23I peptide. As a control, the addition of DMSO was used. The ordinate is the optical density at a wavelength of 600 nm. The abscissa axis is the sampling time.

**Figure 6 ijms-21-06382-f006:**
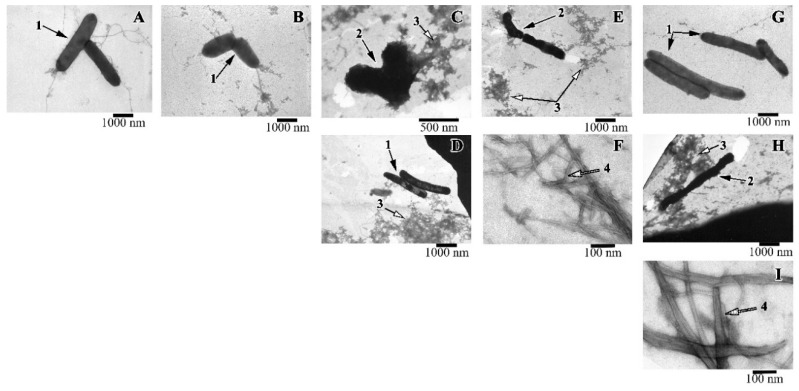
Images of *T. thermophilus* cells after peptide treatment. (**A**): intact cells; (**B**): V10T (100 μg/mL), (**C**): R23I (50 μg/mL); (**D**): R23I (1000 μg/mL); (**E**,**F**): V10I (1000 μg/mL); (**G**): R23T (100 μg/mL); (**H**,**I**): R23T (500 μg/mL). Scale bar is 1000 nm for A, B, D, E, G, H; 500 nm for C and 100 nm for F, I. 1—cells with normal morphology, 2—cells with altered structure, 3—aggregates of proteins and peptides, 4—fibrils.

**Figure 7 ijms-21-06382-f007:**
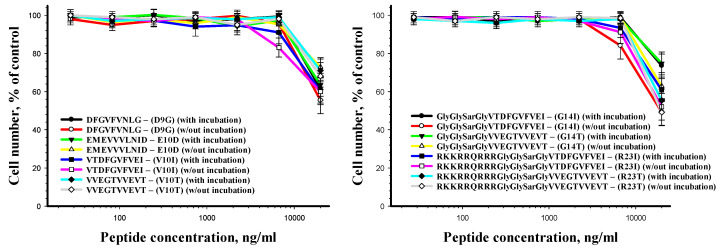
Human fibroblasts viability after peptide treatment as a function of peptide concentration.

**Figure 8 ijms-21-06382-f008:**
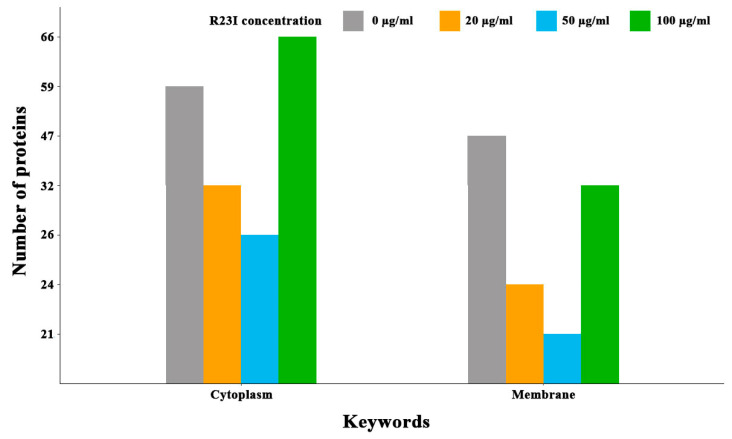
Number of annotated cytoplasmic and membrane proteins depending on the R23I concentration in the cellular environment.

**Figure 9 ijms-21-06382-f009:**
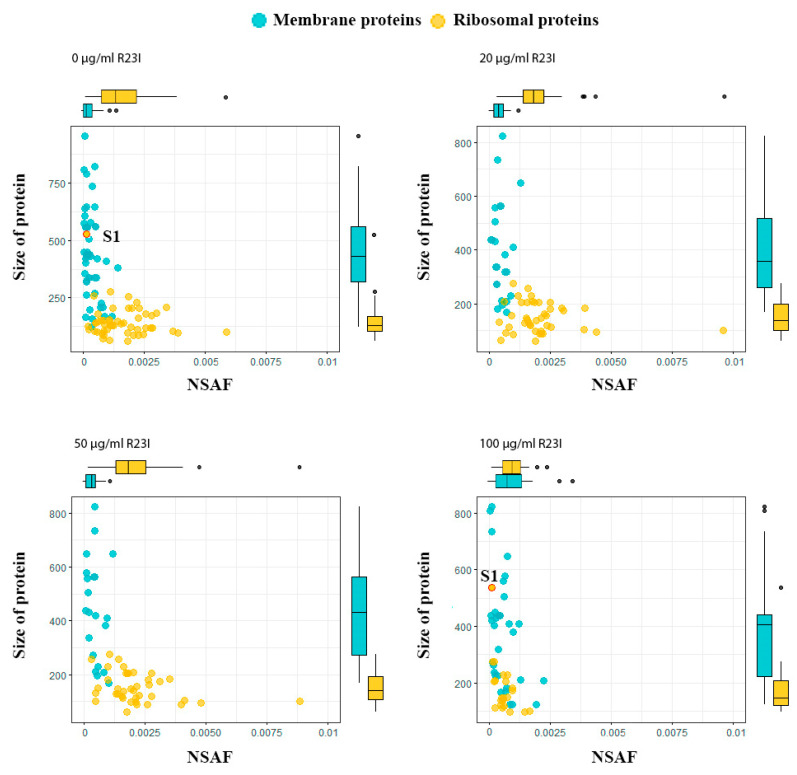
Proteins of *T. thermophilus* cells in the different experimental groups. Boxplots above and on the right describe the distribution of points along the axes. Most of the points lie within the boxplots, outliers are marked with points.

**Table 1 ijms-21-06382-t001:** Antimicrobial prediction for test peptides.

Peptide	Sequence	Mw	pI	CAMPR3 Method			
				Support Vector Machine (svm)	Random Forest (rf)	Artificial Neural Network (ann)	Discriminant Analysis (da)
*D9G*	*DFGVFVNLG*	967.1	3.8	0.00 *	0.41	**	0.03
*E10D*	*EMEVVVLNID*	1160.3	3.4	0.10	0.45	*	0.01
*V10I*	*VTDFGVFVEI*	1125.3	3.5	0.80 **	0.44	*	0.03
*V10T*	*VVEGTVVEVT*	1031.2	3.5	0.08	0.50	*	0.00
*G14I*	*GGSar#(A)GVTDFGVFVEI*	1367.5	3.7	0.134	0.405	*	0.367
*G14T*	*GGSar#(A)G VVEGTVVEVT*	1273.4	3.8	0.04	0.39	*	0.01
*R23I*	*RKKRRQRRRGGSar#(A)GVTDFGVFVEI*	2689.1	12.0	0.03	0.51	*	0.99
*R23T*	*RKKRRQRRRGGSar#(A)GVVEGTVVEVT*	2595.0	12.0	0.01	0.47	*	0.59

*—It is predicted as a peptide that does not exhibit antimicrobial activity. The prediction level is less than 0.5. **—Predicted as an antimicrobial peptide. #—For theoretical calculations alanine was used instead of sarcosine in the synthesized peptides. Molecular weight (Mw) and isoelectric point (pI) of a peptide were calculated by the ExPASy server (https://web.expasy.org/cgi-bin/peptide_mass/peptide-mass.pl) [16].

**Table 2 ijms-21-06382-t002:** Distribution of cells in the phases of the cell cycle.

	G1 Phase	S Phase	G2 Phase	M Phase
Peptides	83 ± 4%	12 ± 3%	5 ± 1%	0%
Control	62 ± 3%	21 ± 3%%	12 ± 2%	5 ± 1%

**Table 3 ijms-21-06382-t003:** Number of detected proteins with the number of identified peptides 3 or more in the studied samples *.

Concentration of R23I, μg/ml	Ribosomal Proteins	All Proteins
0	51 (with S1)	386
20	42 (without S1)	232
50	39 (without S1)	201
100	23 (with S1)	395

*—The condition for homogenization of *T. thermophilus* cells was freezing–thawing.

**Table 4 ijms-21-06382-t004:** Description of some response curves used to describe antimicrobial activity.

Model	Function	Description
Log-logistic model with three parameters (d, b, e)	$\varphi \left(x\right)=\frac{d}{1+e^{b\left(\log(x\right)-\log\left(e\right))}}$	Individual sensitivity is the same for both low and high doses
Log-logistic model with two parameters (b, e)	$\varphi \left(x\right)=\frac{1}{1+e^{b\left(\log(x\right)-\log\left(e\right))}}$	
Exponential model with two parameters (d, e)	$\varphi \left(x\right)=d\left(1-e^{\left(-\frac{x}{e}\right)}\right)$	Description of unknown dependency types
Model Michaelis-Menten with two parameters (d, e)	$\varphi \left(x\right)=\frac{d}{1+\left(\frac{e}{x}\right)}$	Universal hyperbolic dependence of a decrease in growth rate
Weibull model-1 with three parameters	$\varphi \left(x\right)=d\left(e^{e^{b\left(\log(x\right)-\log\left(e\right))}}\right)$	Sensitivity is not the same for low and high concentrations. The Weibull-1 curve slowly decreases in the region of the upper limit, but faster approaches a lower limit

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
