# Peer review of "Antimicrobial and Amyloidogenic Activity of Peptides Synthesized on the Basis of the Ribosomal S1 Protein from Thermus Thermophilus"

_ijms, 2020, doi:10.3390/ijms21176382_

Round 1
Reviewer 1 Report
The manuscript entitled "Antimicrobial and Amyloidogenic Activity of Peptides Synthesized on the Basis of the Ribosomal S1 protein from Thermus thermophilus" coauthored by Stanislav R. Kurpe et al. seems, at first, to be an interesting piece of research. The proposed way of development of microbial peptides might be of value. However, detailed analysis of the results raises multiple questions, predominantly on the novelty of the study and its significance.
Major remarks
The whole section 2.2 "Coagregation of amyloidogenic peptides and ribosomal S1 protein" provides a number of results, however gives no interpretation. What biological implications may those results have? What was the aim of the described measurements? What does it prove? If nothing really, it could be removed from the manuscript.
Regarding peptides R23T and V10I, in my opinion MIC (0.50 and 1.00 mg/ml) is far too high as for a candidate for a potentially therapeutic antibacterial agent. On the other hand, in the case of peptide R23I (0.05 mg/ml) everything would seem right if not for the fact that all the peptides are cytotoxic to human cells above the concentration of 0.02 mg/ml, which is not highlighted by the authors as an issue. For these reasons it might be concluded that none of the peptides is any candidate for an antimicrobial agent.
The section 2.6. "Study of the proteome of intact and peptide-treated T. thermophilus cells", which consumes the whole 6 pages(!) of the manuscript, seems to be highly speculative. Firstly, why is the number of identified proteins so low? We speak here about the whole bacterial proteome. Secondly, is the method properly selected and optimised for making such a comparison? Usually for comparing genomes 2D-DiGE and MS identification of protein spots or MS-based iTRAQ is used. It seems the described differences may be, at least to some extent, a result of poor protein identification based on MS spectra. Thirdly, blocking the function of a crucial protein may entail lots of secondary changes in the proteome, and assigning all the changes to the peptide activity seems dubious. For these reasons I do not find here any significant scientific value.
Minor remarks
Professional language editing would be advised. There are plenty of mistakes and awkward expressions in the manuscript text. To bring a few:
"Understanding the action of peptides that tend to form amyloid can be used as an alternative to antibiotics in the development of antibacterial peptides." Understanding can be used as alternative to antibiotics? Antibiotics in the development of antibacterial peptides?
"the R23I (modified by the peptide of HIV transcription factor for bacterial cell penetration)" How is R23I modified by the peptide of HIV transcription factor?
"It is known that the mechanism of action of most antimicrobial peptides (AMPs) is directed to the cell membrane." Mechanism of action is directed to the cell membrane?
"so it is may be used".
"are capable to significant suppression" capable of?
"The influence of R23I on the ribosomal structural proteins is appeared ambiguously".
"SDS-PAAG"
Concluding, in my opinion the overall quality of the presented results and their interpretation is too low to recommend the manuscript for publication in its current form. I would recommended rethink the approach to the presentation and interpretation of the data, and resubmission.
Author Response
Comments and Suggestions for Authors
The manuscript entitled "Antimicrobial and Amyloidogenic Activity of Peptides Synthesized on the Basis of the Ribosomal S1 protein from Thermus thermophilus" coauthored by Stanislav R. Kurpe et al. seems, at first, to be an interesting piece of research. The proposed way of development of microbial peptides might be of value. However, detailed analysis of the results raises multiple questions, predominantly on the novelty of the study and its significance.
Major remarks
The whole section 2.2 "Coagregation of amyloidogenic peptides and ribosomal S1 protein" provides a number of results, however gives no interpretation. What biological implications may those results have? What was the aim of the described measurements? What does it prove? If nothing really, it could be removed from the manuscript.
Response 1: We thank the reviewer for your effort in critically reviewing our manuscript and for the constructive suggestions and comments. We have provided point-by-point responses to the reviewers’ comments attached herewith.
The selection of peptides was based on their high amyloidogenicity predicted by programs specially designed to search for regions in the protein molecule prone for aggregation. It is known that such sites in various protein and peptide molecules are capable of interacting, which leads to the formation of stable coaggregates (10.1021/acschemneuro.9b00645, 10.1021/acs.nanolett.9b02771) having a hydrophobic core of the corresponding interacting amyloidogenic amino acid residues. In turn, such peptides were developed based on the amyloidogenic sequences of the S1 protein, that was selected as a target for coaggregation. The biological significance of studying peptide coaggregation based on the amyloidogenic sequence of ribosomal S1 protein is as follows. S1 protein is a necessary regulator of translation initiation, elongation, and translation (10.1016/s0022-2836(77)80189-7, 10.1093/emboj/19.23.6612). Moreover, individual domains perform the functions of binding to both other ribosomal proteins and mRNA. Directed coaggregation of regions can have regulatory effects on the important biological processes listed above, which will have a similar gene knockout, but at the protein level. The results of coaggregation are presented in the section "Coagregation of amyloidogenic peptides and ribosomal S1 protein". One can see from the electron microscopic images (Supplementary Tables S1 and S2) that peptides do have a tendency to aggregate with the whole protein. Since coaggregation of the important ribosomal S1 protein will lead to the fact that individual domains cannot fold into a native structure, some important protein functions will be lost, which will have an effect on the growth of T. thermophilus. Exactly the same peptides can be developed on the S1 protein sequence of pathogenic organisms, and then, if successful, they will have an antibacterial effect based on targeted coaggregation.
We have added Figure 1 to clarify the purpose of our research and methods. We are developing peptides based on directed aggregation so that the synthesized peptides have both aggregation and antimicrobial properties.
Regarding peptides R23T and V10I, in my opinion MIC (0.50 and 1.00 mg/ml) is far too high as for a candidate for a potentially therapeutic antibacterial agent. On the other hand, in the case of peptide R23I (0.05 mg/ml) everything would seem right if not for the fact that all the peptides are cytotoxic to human cells above the concentration of 0.02 mg/ml, which is not highlighted by the authors as an issue. For these reasons it might be concluded that none of the peptides is any candidate for an antimicrobial agent.
Response 2: Thank you for your comment. MIC values for promising antibacterial agents can vary significantly depending on the structure of the antibacterial candidate drug itself, experimental conditions, and finally, the characteristics of the bacterial cell should be taken into account. Indeed, at the initial stage of the search for candidates for antimicrobial peptides (AMPs) in vitro, the MIC range can be quite wide and reach the upper values of several mg/ml (10.3390/ani10081340, 10.1038/s41598-020-68951-x). In the future, selected candidate AMP can be modified in such a way as to reduce both the MIC value in relation to the pathogenic organism and the cytotoxic effect in relation to eukaryotic cells (10.3389/fchem.2020.00416, 10.3389/fmicb.2018.02846).
The section 2.6. "Study of the proteome of intact and peptide-treated T. thermophilus cells", which consumes the whole 6 pages(!) of the manuscript, seems to be highly speculative.
Response 3: Thanks, we have shortened the text in the section 2.6.
Firstly, why is the number of identified proteins so low? We speak here about the whole bacterial proteome.
Response 4: The low number of identified proteins is associated with rather strong selection criteria for the PEAKS Studio 7.5 and IdentiPy programs for identified peptides. According to the UniProt database, 2201 unique proteins of Thermus thermophilus are known, of which only 376 have a complete description. This indicates that there are difficulties in identifying and studying the proteins of this organism. We were able to define 1126 unique proteins from Thermus thermophilus, of which only 575 had satisfactory coverage.
In the analysis of cells treated with peptides, the number of cells was significantly less than in standard experiments to determine the gross product of cells.
In our work, we use two programs for analyzing MS spectra: PEAKS Software and IdentiPy server. IdentiPy server is distinguished by the fact that it automatically adjusts the parameters for protein identification (10.1021/acs.jproteome.7b00640). For the analysis of proteomes, we selected proteins with the number of identified peptides 3 or more.
Secondly, is the method properly selected and optimised for making such a comparison? Usually for comparing genomes 2D-DiGE and MS identification of protein spots or MS-based iTRAQ is used. It seems the described differences may be, at least to some extent, a result of poor protein identification based on MS spectra.
Response 5:
We had the task of comparative analysis of predominantly ribosomal proteins. Some of them are found in small quantities in cells. Labeling of cystine residues can lead to protein precipitation (10.1016/B978-012226770-3/10708-3.). And labeling of lysine residues with a dye would most likely lead to non-cleavage by trypsin, which can reduce the identification of proteins (this may also be true for the iTRAQ method). In addition, the dye binds covalently to lysine residues; we are not sure that proteins from Thermus thermophilus reactive to the action of the peptide contain sufficient lysine residues. A comparison of label-based quantification methods (iTRAQ and similar methods) and label-free methods shows that 60-70 percent of the proteins in a sample can be quantified using these methods. Most proteins can be assessed by any method. And the advantage of the label-free quantification method can be to detect a higher percentage of low-concentration proteins (10.1039/c5mb00234f). Using the NSAF quantification parameter allows us to achieve good reproducibility and gives a correct quantification (10.1002/rcm.7829). It is important to note that label-free methods are easy to use and do not require complex and expensive sample preparation. Of course, both methods should be used to fully quantify proteome.
Thirdly, blocking the function of a crucial protein may entail lots of secondary changes in the proteome, and assigning all the changes to the peptide activity seems dubious. For these reasons I do not find here any significant scientific value.
Response 6: Undoubtedly, a change in the concentration of functionally important proteins can lead to a change in the cell proteome. In our study, we did not try to give a complete description of the disappearance and appearance of proteins in the proteome under the action of the peptide. We were more interested in two groups of proteins: membrane and ribosomal. We assumed that the penetration of AMPs into the cell can lead to disruption/change in the functioning of the translation apparatus of the cell. Comparison of these groups allowed us to make an assumption about the mechanism of action of the R23I peptide. This is the basis for further research of the mechanism of action. Therefore, in the article we paid attention to changes in these groups of proteins.
Minor remarks
Professional language editing would be advised. There are plenty of mistakes and awkward expressions in the manuscript text. To bring a few:
"Understanding the action of peptides that tend to form amyloid can be used as an alternative to antibiotics in the development of antibacterial peptides." Understanding can be used as alternative to antibiotics? Antibiotics in the development of antibacterial peptides?
"the R23I (modified by the peptide of HIV transcription factor for bacterial cell penetration)" How is R23I modified by the peptide of HIV transcription factor?
"It is known that the mechanism of action of most antimicrobial peptides (AMPs) is directed to the cell membrane." Mechanism of action is directed to the cell membrane?
"so it is may be used".
"are capable to significant suppression" capable of?
"The influence of R23I on the ribosomal structural proteins is appeared ambiguously".
"SDS-PAAG"
Response 7: Thanks, we have corrected these sentences.
Concluding, in my opinion the overall quality of the presented results and their interpretation is too low to recommend the manuscript for publication in its current form. I would recommended rethink the approach to the presentation and interpretation of the data, and resubmission.
Response 8: Thank you, we found your comments extremely helpful and have revised accordingly.
Reviewer 2 Report
Dear Authors,
It is a very nice, correct manuscript. It is definitely longer than most papers of this Journal. May I suggest the reduction of the size and documentation (tables and figures) of the chapter “2.6 Study of the proteome in intact and peptide-treated T. thermophilus cells”. If you like to publish all information currently presented in this chapter, try to cut it into two or try part, as now it is hard to follow.
Author Response
Comments and Suggestions for Authors
Dear Authors,
It is a very nice, correct manuscript. It is definitely longer than most papers of this Journal. May I suggest the reduction of the size and documentation (tables and figures) of the chapter “2.6 Study of the proteome in intact and peptide-treated T. thermophilus cells”. If you like to publish all information currently presented in this chapter, try to cut it into two or try part, as now it is hard to follow.
Response 1: We would like to thank reviewer for the valuable suggestions. We have changed the text of our manuscript according to the recommendations.
Round 2
Reviewer 1 Report
I would like to thank authors for their valuable explanations. The rationale behind the study as well as the results and their interpretation are presented now in a much clearer and more holistic way without leaving anything important untold and without too much of speculation. Thus, I recommend the manuscript for publication.